# Remapping the spatial distribution of neutralizing sites and their immunodominance on the capsid of different topotypes of FMDV serotype O by site-directed competitive ELISA for detection of neutralizing antibodies

Qiongqiong Zhao,[1,2,3,4] Fengjuan Li,[5] Shulun Huang,[5] Xiangchuan Xing,[5] Ying Sun,[5] Pinghua Li,[5] Huifang Bao,[5] Yuanfang Fu,[5] Pu Sun,[5] Xingwen Bai,[5] Hong Yuan,[5] Xueqing Ma,[5] Zhixun Zhao,[5] Jing Zhang,[5] Jian Wang,[5] Tao Wang,[5] Dong Li,[5] Qiang Zhang,[5] Ping Qian,[1,2,3,4] Yimei Cao,[5] Kun Li,[5] Xiangmin Li,[1,2,3,4] Zengjun Lu[5]

**ABSTRACT** The foot-and-mouth disease virus (FMDV) serotype O contains at least five neutralizing antigenic sites, yet the structural relationship and antibody abundance remain poorly characterized. This study identifies six distinct neutralizing antigenic sites by evaluating 27 host-derived neutralizing antibodies (NAbs) using competitive enzyme-linked immunosorbent assay (cELISA). These sites include the VP1 G-H loop, VP1 C-terminus, site 2, site 4, site 6, and site 7. Notably, classical sites 1 and 5 were reclassified into the VP1 G-H loop and VP1 C-terminus classes. Sites 2 and 4 align with classical classifications, targeting independent epitopes on VP2 and VP3, respectively. We identified two novel sites: site 6, which involves extensive interactions with the G-H loop, C-terminus of VP1 and VP3, and site 7, which interacts with both VP2 and VP3. Sera from cattle, sheep, and pigs immunized with four serotype O lineages (O/SCGH/2016, O/Mya/98, O/Tibet/99, and O/XJ/2017) were used to evaluate the immunodominance of these sites. NAb responses favored site 4 for O/SCGH/2016 and the VP1 G-H loop for O/XJ/2017. Immunization effectiveness varied by strains and host species: O/XJ/2017 and O/Tibet/99 were effective in sheep, while O/Mya/98 showed reduced efficacy; O/Tibet/99 showed good immunogenicity in pigs. No significant differences were observed in cattle. There is a strong correlation (r = 0.8693) between NAb levels at site 6 and virus neutralization tests, suggesting its potential for use in alternative testing methods. This study describes the spatial distribution of neutralizing sites and highlights strain-specific immunodominant epitopes and differential antibody responses across species, providing valuable insights for FMD prevention and control.

**IMPORTANCE** The antigenic structure of the foot-and-mouth disease virus (FMDV) serotype O is complex, and the immunodominant epitopes among different lineages remain poorly understood. This study classified the capsid surface epitopes into six distinct antigenic sites utilizing 27 neutralizing antibodies (NAbs) by paired competitive ELISAs (cELISAs). High-affinity NAbs were selected for site-directed cELISAs to assess antibody abundance in serum from cattle, sheep, and pigs vaccinated with various inactivated FMDV serotype O vaccines. Additionally, liquid-phase blocking ELISA (LPBE) and virus neutralization test (VNT) were employed to measure total antibody and NAb titers. Results indicated that immunodominant sites vary among different strains, with each strain exhibiting different immunogenicity across the three animal species. Notably, antibody titers from NAb pO18-10, targeting site 6 on VP1 and VP3, correlated strongly with VNT results. This study provides comprehensive insights into the antigenic structure

**Peer Reviewers** Kelsey A. Pilewski, US Food and Drug Administration, Silver Spring, Maryland, USA; Shawn Babiuk, Canadian Food Inspection Agency, Winnipeg, Canada

Address correspondence to Xiangmin Li, lixiangmin@mail.hzau.edu.cn, Yimei Cao, caoyimei@caas.cn, Kun Li, likun02@caas.cn, or Zengjun Lu, luzengjun@caas.cn.

The authors declare no conflict of interest.

See the funding table on p. 17.

of FMDV serotype O and lays the groundwork for developing new methods to detect NAbs.

**KEYWORDS**   foot-and-mouth disease virus, serotype O, neutralizing antigenic sites, neutralizing antibody, competitive ELISA

Foot-and-mouth disease (FMD) is a highly contagious disease threatening cloven-hoofed animals such as cattle, sheep, goats, pigs, and deer (1). It is recognized as one of the most economically significant livestock diseases worldwide, with a global distribution except in North America, Western Europe, and Australia (2). The etiological agent of FMD is the foot-and-mouth disease virus (FMDV), classified within the genus *Aphthovirus* of the family *Picornaviridae*, featuring a single-stranded positive RNA genome (3). FMDV encompasses seven antigenically distinct serotypes: A, O, C, Asia1, SAT1, SAT2, and SAT3, each displaying multiple lineages or topotypes based on their endemic regions (4). Notably, serotype O encompasses 11 genetically distinct topotypes and a variety of genotypes, and accounts for over 60% of global outbreaks, posing significant challenges for the prevention and control of FMD (2, 5). Recent predominant lineages of FMDV serotype O circulating in Asia include O/SEA/Mya98, O/Middle East-South Asia (ME-SA)/PanAsia, O/ME-SA/India2001, and O/Cathay (6). Due to the high variability of the virus, limited cross-protection is observed among different topotypes or strains (7–9). So, vaccine selection and matching tests are essential operations in response to new viral incursions when vaccination strategies are employed for disease control.

Currently, vaccination with inactivated vaccines remains the primary strategy for FMD control in endemic regions (10). A strong correlation had been established between neutralizing antibodies (NAbs) and protective immunity across species (11). The virus neutralization test (VNT) is regarded as the gold standard for detecting FMDV-specific NAbs, noted for its exceptional sensitivity and specificity, making it the preferred diagnostic method for vaccine matching and assessing protective immune responses *in vitro* (12). However, VNT is labor-intensive and time-consuming, highlighting the need for alternative methods to accurately determine NAb levels. Our previous study demonstrated that a bovine-derived NAb W145 targeted to antigenic site 5 (13) can be used to develop a competitive ELISA (cELISA) for the detection of NAbs against serotype A with good correlation to VNT (14). An earlier study in our laboratory showed that two NAbs, E46 and F128, targeted to VP2 and VP3 of serotype O (15) were used to develop a double blocking ELISA for the detection of NAbs against serotype O with less well correlation to VNT (16). These results imply that we need to determine the immunodominant epitopes inducing NAbs responses and clarify the spatial relationship of different neutralizing antigenic sites, which will provide new clues for accurate detection of NAbs upon different virus antigen vaccinations.

The mature FMDV particle comprises 60 copies of its four structural proteins encoded as VP1, VP2, VP3, and VP4 (17). Previous studies have identified five classical neutralizing antigenic sites on the surface of FMDV serotype O (18–20). Antigenic site 1 is linear and sensitive to trypsin, while the remaining sites are conformation dependent and trypsin resistant. Antigenic site 1 involves the G-H loop and C-terminus of VP1, with critical residues located at positions 144, 148, 154, and 208. The key residues for antigenic site 2 are at positions 70-73, 75, 77, 131, and 134 of VP2. Antigenic site 3 involves critical residues at positions 43 and 44 on the VP1 B-C loop. Amino acid residues at positions 56 and 58 of VP3 have been identified as critical for antigenic site 4. Antigenic site 5 likely results from interactions between the VP1 G-H loop and other surface amino acids, with a significant role attributed to the key amino acid at position 149 of VP1 (19–23). Recently, the widespread use of cryo-electron microscopy has significantly promoted the study of the conformational epitopes of FMDV serotype O. Analysis of the complex structure of FMDV serotype O with a serotype O/A cross-reacting NAb (R50) revealed a novel neutralizing antigenic site involving residues from the VP1 B-C loop (50 and 52), E-F loop (94-95), G-H loop (157, 159, and 160), and VP3 G-H loop (173 and 177) (24).

A novel set of epitopes was determined by C4 and C4-like antibodies binding with key amino acids in B-B knob (65), B-C loop (68), E-F loop (131 and 134), and H-I loop (196) of VP3 (15). Additionally, the single-domain NAb M8, derived from llamas and capable of cross-neutralizing serotypes O, A, Asia1, and C, recognizes an epitope that includes residues in both VP1 (positions 93-95 and 98 of the E-F loop, and 133, 143-144, 147-149, 151-153, 161, 164, and 167 of the G-H loop) and VP3 (positions 171-172 and 175-177 of the G-H loop). An O-specific NAb, M170, targets a conformational epitope comprising multiple residues in VP3 (positions 57-59 of the B-B knob, 71 and 73 of the B-C loop, 78 and 85 of the C-D loop, 131 and 134 of the E-F loop, and 177-178 and 183 of the G-H loop) as well as residues 199-202 at the C-terminus of VP1 (25). These results indicate the complexity of antigenic sites on the capsid of FMDV serotype O.

Variations in the critical residues of the neutralizing sites can lead to dynamic changes in the predominant epitopes within the antigen and further affect the efficacy of cross-protection of topotypes or lineages of FMDV serotype O. The RGD motif in VP1 G-H loop serves as a binding site for integrin receptors and plays a crucial role in viral invasion of host cells. This region is also a highly variable structural domain containing multiple epitopes responsible for virus neutralization, such as antigenic sites 1 and 5 (19, 21). Historically, antigenic site 1 has been recognized as immunodominant (26–28). Peptide vaccines corresponding to VP1 residues (141-160 and 200-213) have demonstrated the induction of NAbs and protective effects in guinea pigs and cattle following immunization (29–31). A monoclonal antibody (mAb)-based cELISA revealed that sites 1, 2, and 3 did not exhibit clear immunodominance in polyclonal sera from cattle, sheep, and pigs (32). Furthermore, analysis using a panel of mAb-neutralization escape mutant viruses revealed that antigenic site 2 emerged as the predominant immunogenic site responsible for provoking NAbs production against FMDV serotype O, followed by antigenic site 1 (33). However, employing the same method showed that polyclonal sera from primary-vaccinated cattle exhibited significantly higher levels of antibodies to antigenic site 2, whereas both antigenic sites 1 and 2 demonstrated equal importance in polyclonal sera from multiply vaccinated animals (34).

To our knowledge, no comprehensive study has yet estimated the relative importance of different neutralizing antigenic sites in antisera from cattle, sheep, and pigs conventionally immunized with four lineages within three topotypes of FMDV serotype O (O/Mya/98 of SEA topotype, O/SCGH/2016 of Cathay topotype, O/Tibet/99, and O/XJ/2017 of ME-SA topotype). This study employs a series of bovine- or porcine-derived single B-cell NAbs to develop cELISAs for detection of the NAb abundance to different antigenic sites. Initially, 27 antibodies reactive to serotype O strains were categorized to target six structurally distinct antigenic sites using cELISA. Subsequently, six high-affinity representative antibodies were selected to develop site-directed cELISA: these are B66 targeting to VP1 C-terminus, pOA-1 to VP1 G-H loop, pO18-12 to antigenic site 2 in VP2, pO18-11 to site 4 in VP3, pO18-10 binding to VP1 and VP3 (designated as site 6), and B82 binding to VP2 and VP3 (designated as site 7). These site-directed cELISAs were used to evaluate the antibody level to different antigenic sites by detection of sera collected from cattle, sheep, and pigs vaccinated with four lineages of FMDV serotype O antigens. Additionally, liquidphase blocking ELISA (LPBE) and VNT were employed to measure total antibody and NAb titers. This study revealed immunodominant sites and their variation among different topotypes of FMDV serotype O and across major hosts. These findings will inform the selection of vaccine strains and the design of new vaccine antigens.

## RESULTS

### Selection of ELISA-reactive bovine or porcine NAbs and development of site-directed cELISA

In our previous work, we utilized single B-cell antibody technology to generate a series of broadly NAbs against FMDV serotype O from natural host cattle and pigs (15, 35, 36). From these NAbs, we selected 35 antibodies that showed good reactivity in

ELISA, and their binding epitopes were primarily identified by screening neutralization-resistant mutants (Table S1). The 35 antibodies were classified according to the five classical antigenic sites on FMDV serotype O (Table S2). Specifically, 10 NAbs (pOA-6, pOA-13, pOA-20, pO18-8, pO18-10, pO18-17, pO18-52, pO18-53, pO18-57, and B66) target antigenic site 1, with critical residues located at positions 148 in the G-H loop or 204, 207 in the C-terminus of VP1. Eleven NAbs (B57, B73, B77, B82, F28, pOA-2, pO18-12, pO18-16, pO18-20, pO18-24, and pO18-26) target antigenic site 2 involving the critical residues at positions 68, 71, 72, 77, 188, 190, and 195 in VP2. Four NAbs (A19, B74, C5, and E18) recognize classical antigenic site 3 with critical residues at positions 43 and 58 in VP1. Moreover, six NAbs (C4, F169, pO18-2, pO18-11, pO18-28, and pO18-54) bind to classical antigenic site 4 with key amino acids at positions 65, 68, 70, 76, 131, 134, 174, and 209 in VP3. Finally, four NAbs (pOA-1, pOA-7, pO18-39, and pOA-40) recognize classical antigenic site 5 with critical amino acids at position 149 in VP1.

Subsequently, we try to develop site-directed cELISA by using these 35 NAbs. Based on checkerboard titration results, one cross-reactive non-neutralizing mAb E32 (optimal concentration to be 0.5 µg/mL) was used to capture the whole virus 146S antigen of FMDV serotype O at a concentration of 1 µg/mL. The procedure of cELISA was similar to a previous publication (14). The optimal reaction conditions for the site-directed cELISA are listed in Table S3. E32 was used to capture O/Mya/98 and O/Tibet/99 antigens, and an indirect ELISA was employed to determine the working concentrations of 35 biotinylated NAbs (Biotin-NAbs) for cELISA to ensure result readability. Initially, the 35 Biotin-NAbs were set at a concentration of 5 µg/mL and then diluted twofold to select a dilution with an absorbance at 450 nm (A450) value of approximately 2.0 for cELISA. All 35 NAbs were subsequently tested in a self-cELISA, with 27 demonstrating effective self-competitive capability (Fig. 1A). The working concentrations of these 27 antibodies are detailed in Table S4. However, eight antibodies showed self-competitive inhibition rates of either 0 or approximately 50%, including three from site 1 (pO18-52, pO18-53, and pO18-57), one from site 2 (B77), and all four from site 3 (A19, B74, C5, and E18). The working concentrations and self-competitive inhibition rates for these antibodies are provided in Table S5.

## Reclassification of the antigenic sites on FMDV serotype O based on pairwise cELISA results of 27 NAbs

Twenty-seven NAbs were used to perform pairwise cELISA using E32 to capture O/Mya/98 (Fig. 1A) and O/Tibet/99 (Fig. S1) antigens, respectively, as a repeat. According to the inhibition rates of cELISA, we can divide 27 NAbs into six groups recognizing different antigenic sites, which are VP1 G-H loop, VP1 C-terminus, site 2, site 4, site 6, and site 7. Despite the diverse antigenic profiles of O/Mya/98 and O/Tibet/99, the classification of antigenic sites using 27 NAbs in cELISA remained consistent. The NAb B66, which specifically binds to the VP1 C-terminus, did not exhibit complete competition with NAbs targeting the VP1 G-H loop (pOA-1, pO18-39, pO18-40, pOA-7, pOA-20, pOA-13, and pOA-6), suggesting that the VP1 C-terminus and G-H loop are spatially independent and likely represent different antigenic sites. Meanwhile, B66 showed a relatively higher competitive rate with pO18-54 binding to VP3 C-terminus than those binding to VP1 G-H loop in cELISA, which indicated both VP1 and VP3 C-terminus were spatially adjacent. NAbs targeted to a linear epitope in VP1 G-H loop (pOA-1, pOA-6, pOA-7, pOA-13, and pOA-20) (36) and an epitope with critical amino acids 149 of VP1 G-H loop (pO18-39 and pO18-40) were categorized to be one antigenic site (designated as VP1 G-H loop class) based on the mutual competition between these NAbs. Two NAbs, pO18-10 and pO18-17, showed a high competitive rate in cELISA with NAbs binding to the G-H loop (pOA-1, pOA-6, pOA-7, pOA-13, pOA-20, pO18-39, and pO18-40) or C-terminus of VP1 (B66) or VP3 (pO18-2, pO18-8, pO18-28, and pO18-54), indicating these two NAbs recognize a new conformational antigenic site (designated as site 6) involving multiple critical residues on VP1 G-H loop, C-terminus, and VP3. The epitopes recognized by NAbs B73, pOA-2, and pO18-12 are primarily located on VP2 and are therefore classified into

**A**

**Inhibition of biotinylated NAb binding (%)**

Competitor NAb columns across the top; Biotin-NAb rows on the left.

| Biotin-NAb \ Competitor NAb | pOA-1 | pO18-39 | pO18-40 | pOA-7 | pOA-20 | pOA-13 | pOA-6 | B66 | pO18-10 | pO18-17 | B57 | F28 | B82 | pO18-24 | pO18-20 | pO18-26 | pO18-16 | B73 | pOA-2 | pO18-12 | C4 | F169 | pO18-2 | pO18-28 | pO18-8 | pO18-54 | pO18-11 | Antigenic site (Host-derived) | Antigenic site (Murine-derived) |
|---|---|---|---|---|---|---|---|---|---|---|---|---|---|---|---|---|---|---|---|---|---|---|---|---|---|---|---|---|---|
| pOA-1 | 100 | 100 | 100 | 100 | 92 | 94 | 86 | 39 | 39 | 26 | 32 | 13 | 17 | 15 | 14 | 21 | 16 | 21 | 45 | 38 | 17 | 22 | 37 | 17 | 24 | 48 | 26 | VP1 G-H loop | Site 5 |
| pO18-39 | 101 | 101 | 101 | 101 | 97 | 98 | 91 | 43 | 39 | 26 | 33 | 16 | 19 | 19 | 16 | 15 | 12 | 15 | 47 | 42 | 15 | 23 | 37 | 22 | 28 | 42 | 15 | | Site 5 |
| pO18-40 | 101 | 100 | 100 | 101 | 97 | 98 | 91 | 32 | 18 | 10 | 13 | 6 | 12 | 14 | 7 | 15 | 7 | 14 | 44 | 31 | 13 | 16 | 18 | 8 | 15 | 40 | 6 | | Site 5 |
| pOA-7 | 100 | 100 | 100 | 101 | 93 | 93 | 86 | 18 | 18 | 7 | 11 | 6 | 10 | 15 | 4 | 13 | 8 | 7 | 31 | 25 | 8 | 10 | 31 | 11 | 17 | 36 | 10 | | Site 5 |
| pOA-20 | 102 | 101 | 100 | 102 | 102 | 102 | 97 | 39 | 53 | 31 | 21 | 3 | 1 | -3 | 3 | 15 | 4 | 14 | 40 | 19 | 0 | 10 | 34 | 10 | 17 | 69 | 10 | | Site 1 |
| pOA-13 | 101 | 101 | 100 | 101 | 100 | 101 | 92 | 36 | 42 | 26 | 19 | 16 | 21 | 21 | 9 | 20 | 9 | 19 | 41 | 26 | 17 | 6 | 18 | 10 | 14 | 64 | 6 | | Site 1 |
| pOA-6 | 100 | 100 | 99 | 100 | 101 | 100 | 100 | 69 | 72 | 62 | 27 | 11 | 7 | 18 | 10 | 19 | 12 | 20 | 56 | 29 | 8 | 31 | 57 | 26 | 36 | 89 | 6 | | Site 1 |
| B66 | 8 | 12 | 12 | 3 | 12 | 14 | 12 | 100 | 37 | 25 | 13 | 15 | 18 | 16 | 19 | 22 | 17 | 14 | 29 | 32 | 24 | 22 | 29 | 20 | 20 | 78 | 23 | VP1 C-terminus | |
| pO18-10 | 92 | 76 | 81 | 88 | 96 | 92 | 98 | 100 | 97 | 92 | -38 | -53 | -39 | -43 | -41 | -27 | -43 | -37 | 48 | 35 | -56 | -52 | 99 | 100 | 100 | 99 | -65 | | Site 6 |
| pO18-17 | 82 | 67 | 76 | 72 | 90 | 86 | 96 | 99 | 93 | 90 | -17 | -16 | -6 | -23 | -31 | -8 | -27 | -11 | 59 | 45 | -38 | -44 | 98 | 98 | 98 | 96 | -49 | | Site 6 |
| B57 | 64 | 77 | 81 | 64 | 58 | 58 | 48 | -25 | -23 | -27 | 92 | 96 | 96 | 99 | 98 | 100 | 99 | 64 | 99 | 100 | 100 | 99 | 99 | 100 | 101 | 52 | -11 | | Site 7 |
| F28 | 42 | 36 | 22 | 3 | 14 | -37 | 2 | 18 | -11 | -3 | 88 | 87 | 94 | 95 | 91 | 95 | 92 | 43 | 95 | 96 | 98 | 98 | 97 | 99 | 98 | 45 | -5 | | Site 7 |
| B82 | 23 | 39 | 49 | 32 | -21 | 23 | -10 | -1 | -12 | -8 | 86 | 95 | 101 | 102 | 100 | 102 | 100 | 80 | 102 | 102 | 99 | 100 | 95 | 95 | 98 | 13 | 24 | | Site 7 |
| pO18-24 | 4 | 19 | 25 | 15 | 1 | 2 | -2 | -4 | -1 | -7 | 73 | 86 | 98 | 101 | 98 | 101 | 97 | 57 | 93 | 99 | 95 | 95 | 92 | 98 | 98 | -11 | -4 | | Site 7 |
| pO18-20 | 11 | 14 | 34 | 19 | 1 | 13 | -20 | -6 | 1 | -6 | 80 | 89 | 97 | 99 | 101 | 102 | 99 | 67 | 100 | 101 | 100 | 99 | 99 | 101 | 102 | 6 | 8 | | Site 7 |
| pO18-26 | 8 | 23 | 37 | 23 | 11 | 11 | 5 | 8 | 7 | 5 | 75 | 85 | 98 | 101 | 98 | 101 | 97 | 62 | 98 | 99 | 96 | 96 | 93 | 97 | 98 | -2 | 4 | | Site 7 |
| pO18-16 | -8 | -7 | 4 | -8 | -9 | 4 | -2 | 6 | -8 | 3 | 30 | 44 | 81 | 100 | 92 | 100 | 90 | 14 | 92 | 97 | 90 | 89 | 76 | 86 | 93 | -1 | -2 | | Site 7 |
| B73 | 53 | 55 | 63 | 55 | 20 | 49 | 25 | 14 | 4 | -11 | 83 | 78 | 85 | 85 | 84 | 82 | 84 | 82 | 88 | 88 | 93 | 92 | -54 | -36 | -48 | 36 | 2 | | Site 2 |
| pOA-2 | 10 | 21 | 23 | 19 | -4 | 8 | -5 | 5 | 4 | -8 | 82 | 90 | 99 | 99 | 98 | 96 | 96 | 62 | 100 | 100 | 97 | 58 | 22 | 16 | -1 | 10 | 28 | | Site 2 |
| pO18-12 | -4 | 22 | 41 | 17 | -22 | -5 | -28 | -3 | -2 | -4 | 82 | 88 | 99 | 96 | 97 | 91 | 96 | 27 | 99 | 100 | 75 | 29 | -12 | 2 | 10 | -4 | 20 | | Site 2 |
| C4 | -2 | 12 | 27 | 8 | 2 | 9 | 10 | 13 | 4 | 0 | 43 | 49 | 63 | 59 | 56 | 73 | 58 | 48 | 76 | 75 | 100 | 99 | 96 | 98 | 100 | 91 | 50 | | Site 4 |
| F169 | -2 | 15 | 28 | 11 | 5 | 16 | 9 | 18 | 5 | -2 | 28 | 31 | 46 | 40 | 36 | 53 | 34 | 27 | 55 | 36 | 97 | 99 | 97 | 100 | 100 | 87 | 50 | | Site 4 |
| pO18-2 | 3 | 13 | 21 | 3 | 0 | -6 | -1 | 11 | 47 | 25 | 21 | 25 | 36 | 40 | 38 | 53 | 35 | 19 | 17 | 18 | 95 | 97 | 100 | 100 | 101 | 96 | 49 | | Site 4 |
| pO18-28 | -7 | -7 | 5 | -9 | 3 | -10 | -9 | -4 | 19 | 11 | 1 | -1 | 9 | 22 | 10 | 34 | 13 | 0 | -4 | 1 | 92 | 95 | 97 | 100 | 100 | 90 | 18 | | Site 4 |
| pO18-8 | -14 | 0 | 7 | -16 | -11 | -11 | -4 | 4 | 40 | 28 | 15 | 5 | 13 | 31 | 27 | 46 | 28 | 15 | 16 | 12 | 91 | 93 | 97 | 99 | 101 | 94 | 36 | | Site 4 |
| pO18-54 | 14 | 0 | 9 | 12 | 13 | 24 | 22 | 42 | 51 | 38 | 15 | 13 | 15 | 13 | 0 | 12 | 1 | 22 | 32 | 25 | 88 | 88 | 91 | 94 | 93 | 92 | 46 | | Site 4 |
| pO18-11 | 6 | 17 | 38 | 3 | -13 | -8 | -14 | 12 | -22 | -22 | -4 | -7 | 21 | 11 | 12 | 24 | 17 | 29 | 42 | 43 | 95 | 96 | 98 | 100 | 99 | 83 | 96 | | Site 4 |

**B**

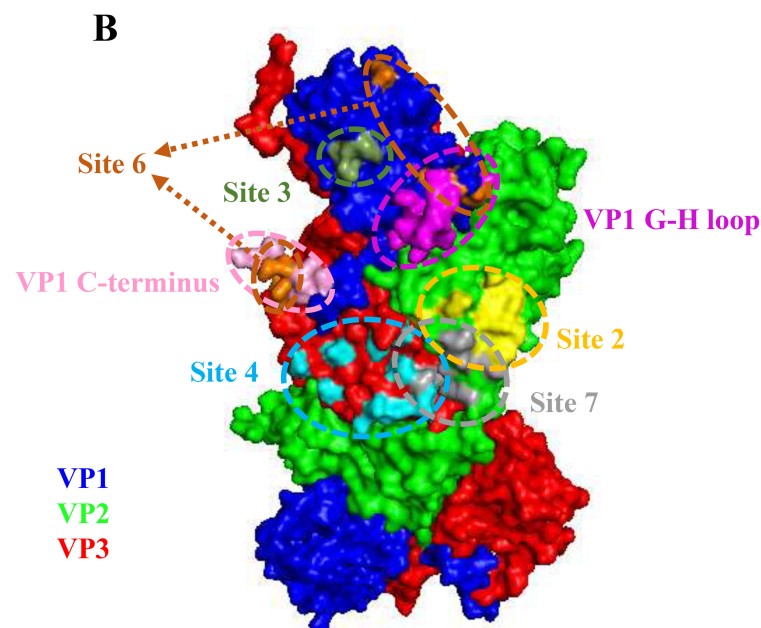

Site 6 Site 3 VP1 G-H loop VP1 C-terminus Site 2 Site 4 Site 7

VP1
VP2
VP3

**FIG 1** Reclassification of antigenic sites and their key determinants on the surface of FMDV serotype O. (A) E32 captures the O/Mya/98 antigen for pairwise cELISA and categorizes 27 host-derived antibodies into six classes based on their competitive relationships. A gradually enhanced red color was used to visualize an increasing percentage inhibition rate of Biotin-NAbs binding. (B) The spatial distribution of seven distinct antigenic sites is plotted on two adjacent pentamers of FMDV. The Prorein Data Bank (PDB) ID for the structure file used to create Figure 1B is PDB:1FOD. The key residues of the epitope are shown in Table 1, drawn as colored spheres on FMDV capsid surface using PyMOL. VP1 G-H loop, magenta; VP1-C terminus, pink; Site 2, yellow; Site 3, atrovirens; Site 4, light blue; Site 6, orange; Site 7, gray; VP1, blue; VP2, green; and VP3, red.

the site 2 class. Additionally, seven NAbs (B57, F28, B82, pO18-24, pO18-20, pO18-26, and pO18-16) showed strong competition with NAbs binding to site 2 and site 4, which indicated another new antigenic site (designated as site 7). Classical antigenic site 4 on VP3 was characterized by seven antibodies (C4, F169, pO18-2, pO18-28, pO18-8, pO18-54, and pO18-11) with the difference that the binding epitopes for C4 and F169 were closer to antigenic site 2, shown by the partial competitive rate with antibodies binding to VP2 in cELISA. Cryo-electron microscopy (Cryo-EM) of the FMDV-OTi (O/Tibet/99)-C4 complex revealed C4 binding to a novel interprotomer antigen epitope around the icosahedral threefold axis of the virus particles including multiple sites in VP2 and VP3 (15), which was further proved by moderate competition between site 2 and site 7 class antibodies with C4 NAb. Interestingly, the binding sites for NAbs pO18-28 and pO18-8 may involve VP1 at position 108; however, these antibodies did not show competitive interactions with NAbs targeting VP1 in the cELISA (Fig. 1A; Fig. S1; Table 1).

In summary, seven different antigenic sites were observed by the results of site-directed cELISA including antigenic site 3, for which we did not find NAbs suitable for cELISA. Figure 1B shows the spatial distribution of the seven antigenic sites by mapping them on the surface of two adjacent protomers of FMDV. These sites are listed as the VP1 G-H loop, VP1 C-terminus, site 2, site 3, and site 4 structurally separated, while site 6 and site 7 exhibit broader contacts for their NAbs binding to multiple sites in VP1 and VP3, VP2 and VP3, respectively.

**TABLE 1** Critical residues of antigenic sites targeted by 27 host-derived monoclonal neutralizing antibodies used in this study

| Antigenic site (Murine derived) | Antigenic site (Host derived) | Host | NAb | Critical residues |
|---|---|---|---|---|
| Site 5 | VP1 G-H loop | Pig | pOA-1 | VP1-133, 138, 149 |
| | | Pig | pO18-39 | VP1-143, 149 |
| | | Pig | pO18-40 | VP1-149 |
| | | Pig | pOA-7 | VP1-149 |
| Site 1 | | Pig | pOA-20 | VP1-148 |
| | | Pig | pOA-13 | VP1-133, 138, 148 |
| | | Pig | pOA-6 | VP1-133, 137, 138, 148 |
| | VP1 C-terminus | Cattle | B66 | VP1-207, 209, 211 |
| Site 6 | | Pig | pO18-10 | VP1-99, 143, 158, 204; VP3-173, 174 |
| | | Pig | pO18-17 | VP1-99, 204 |
| Site 7 | | Cattle | B57 | VP2-71, 72 |
| | | Cattle | F28 | VP2-71, 72 |
| | | Cattle | B82 | VP2-71, 72 |
| | | Pig | pO18-24 | VP2-188, 190, 195 |
| | | Pig | pO18-20 | VP2-71, 72, 188, 190, 195; VP3-197 |
| | | Pig | pO18-26 | VP2-71, 195 |
| | | Pig | pO18-16 | VP2-71, 188, 195; VP3-196, 197, 209 |
| Site 2 | | Cattle | B73 | VP2-72 |
| | | Pig | pOA-2 | VP2-68 |
| | | Pig | pO18-12 | VP2-68, 77, 196 |
| Site 4 | | Cattle | C4 | VP3-65, 68, 69, 131, 134, 195, 196 |
| | | Cattle | F169 | VP3-68 |
| | | Pig | pO18-2 | VP3-70, 76, 131, 134; VP1-199 |
| | | Pig | pO18-28 | VP3-58, 61, 70, 76; VP1-108 |
| | | Pig | pO18-8 | VP1-108, 204 |
| | | Pig | pO18-54 | VP3-174, 209; VP1-194, 204 |
| | | Pig | pO18-11 | VP3-131 |

# Evaluation of antibody abundance to each of six neutralizing sites after immunization of three susceptible animals with four lineages of FMDV serotype O whole-virus antigens

Following the analysis of spatial hindrance effects using site-directed cELISA with 27 NAbs, six representative NAbs were selected to assess antibody abundance at various antigenic sites after the immunization of three natural hosts with four lineages of FMDV serotype O whole-virus inactivated antigens. The chosen representative NAbs were Biotin-B66 (VP1 C-terminus), Biotin-pOA-1 (VP1 G-H loop), Biotin-pO18-12 (site 2), Biotin-pO18-11 (site 4), Biotin-pO18-10 (site 6), and Biotin-B82 (site 7). Notably, VP1 C-terminus, VP1 G-H loop, site 2, and site 4 exhibited spatial independence, allowing for a primary analysis of antibody immunodominance at these sites. NAb level in serum samples was quantified by determining the maximum serum dilution that inhibits at least 50% of NAb binding in cELISA. Total NAbs were detected by VNT at 49 days post-vaccination (dpv) after booster immunization. Figure 2 to 4 shows the results of site-directed cELISA for NAbs to different sites after vaccination of cattle, sheep, and pigs with four lineages of serotype O antigens. The O/Mya/98 antigen was used in cELISA to detect the NAbs in serum samples collected at 21 and 49 dpv after primary and boost vaccinations (Fig. 2 to 4), while O/Tibet/99 antigen was used in cELISA to test serum samples collected at 49 dpv as a comparison (Fig. S2). Despite the use of different topotype antigens in cELISA, the results were remarkably similar between the two antigens. Summary statistics for antibody values (log10) obtained from site-directed cELISAs, LPBE, and VNT are provided in Tables S6 to S8. A one-way analysis of variance

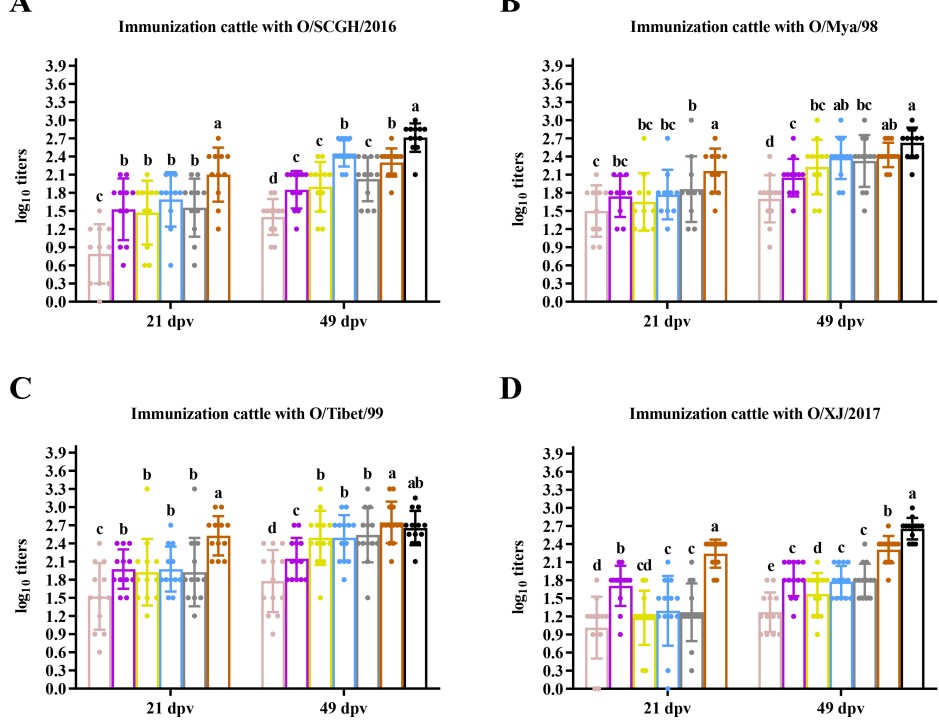

FIG 2 Comparison of antibody abundance against various neutralizing sites after immunization of cattle with four lineages of FMDV serotype O vaccines. The E32 antibody was utilized to capture the O/Mya/98 antigen, with cattle serum samples collected at 21 and 49 dpv after primary and booster immunization with the O/SCGH/2016 (A), O/Mya/98 (B), O/Tibet/99 (C), and O/XJ/2017 (D) vaccines. Antibody abundances were assessed using a site-directed cELISA targeting six different epitopes: Biotin-B66 (VP1 C-terminus), Biotin-pOA-1 (VP1 G-H loop), Biotin-pO18-12 (VP2), Biotin-pO18-11 (VP3), Biotin-B82 (VP2 and VP3), and Biotin-pO18-10 (VP1 and VP3). Total NAb titers to O/Mya/98 virus at 49 dpv were determined through a VNT test. Error bars represent 95% confidence intervals. A one-way ANOVA, followed by Tukey's multiple comparisons test, was conducted to assess the statistical significance of differences in antibody abundance across various sites within the same lineage. Data points from sera collected at 21 and 49 dpv are annotated with different letters to denote statistically significant differences ($P < 0.05$). The marked alphabets (a to e) indicate the ranking of titers from high to low for different groups.

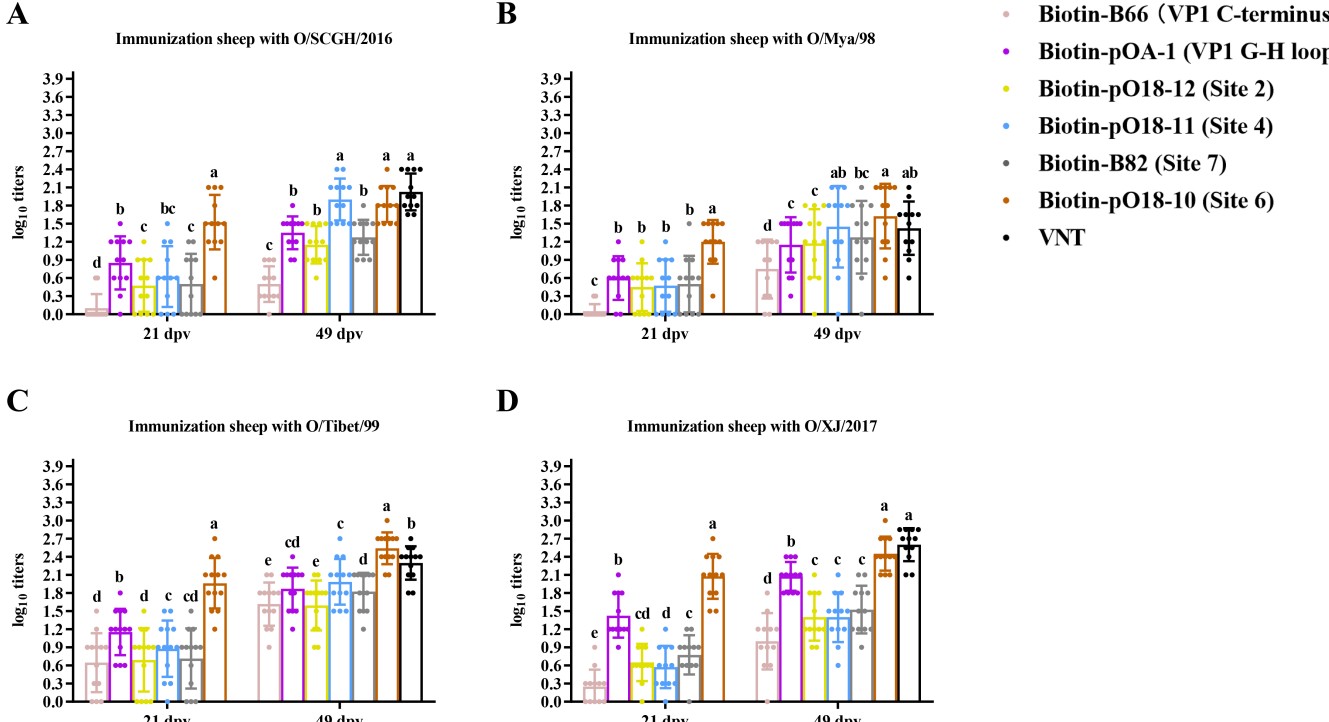

**FIG 3** Comparison of antibody abundance against various neutralizing sites after immunization of sheep with four lineages of FMDV serotype O vaccines. The same cELISA was used to detect sheep serum samples collected at 21 and 49 dpv after immunization with the O/SCGH/2016 (A), O/Mya/98 (B), O/Tibet/99 (C), and O/XJ/2017 (D) vaccines. Antibody abundance was assessed using a site-directed cELISA targeting six epitopes described in Figure 2. Total NAb titers to O/Mya/98 virus at 49 dpv were determined through a VNT test. A one-way ANOVA (Tukey's multiple comparisons test) was the same as those described in Figure 2.

(ANOVA) (Tukey's multiple comparisons test) was employed to examine differences in antibody abundance across the six sites in cELISA at 21 dpv, as well as differences between cELISA and VNT results at 49 dpv. Significant differences are indicated by different letters, where the same letter denotes no significant difference ($P > 0.05$), while different letters indicate significant differences ($P < 0.05$). Variations in antibody responses to different sites among the three animal species were evident following vaccination with distinct virus antigens. Overall, responses to the VP1 C-terminus (probed by Biotin-B66) were the lowest compared to other sites across all vaccinated animals, whereas mean NAb titers probed by Biotin-pO18-10 (site 6) were the highest among the three species.

Cattle exhibited robust antibody responses after vaccination with four different types of O antigens. At 21 dpv, no significant differences were observed in antibody responses to the VP1 G-H loop, site 2, and site 4 when vaccinated with O/SCGH/2016, O/Mya/98, and O/Tibet/99 antigens ($P > 0.05$) (Fig. 2A through C). Conversely, cattle vaccinated with O/XJ/2017 demonstrated significantly higher antibody responses to the VP1 G-H loop compared to the VP1 C-terminus, site 2, and site 4 ($P < 0.05$), indicating the immunodominance of this site for O/XJ/2017 (Fig. 2D). Following booster vaccination, all cattle demonstrated increased antibody responses (Fig. 2A through D). Specifically, in cattle vaccinated with O/SCGH/2016, antibody levels at 49 dpv for site 4 were significantly higher than those for the VP1 C-terminus, VP1 G-H loop, and site 2 ($P < 0.05$), confirming the immunodominance of site 4 for this antigen (Fig. 2A; Fig. S2A). For cattle vaccinated with O/Mya/98, O/Tibet/99, and O/XJ/2017 antigens at 49 dpv, no obvious immunodominance sites were observed (Fig. 2B through D; Fig. S2B through D). However, the relative average distribution of the antibody responses to these sites was obvious with the

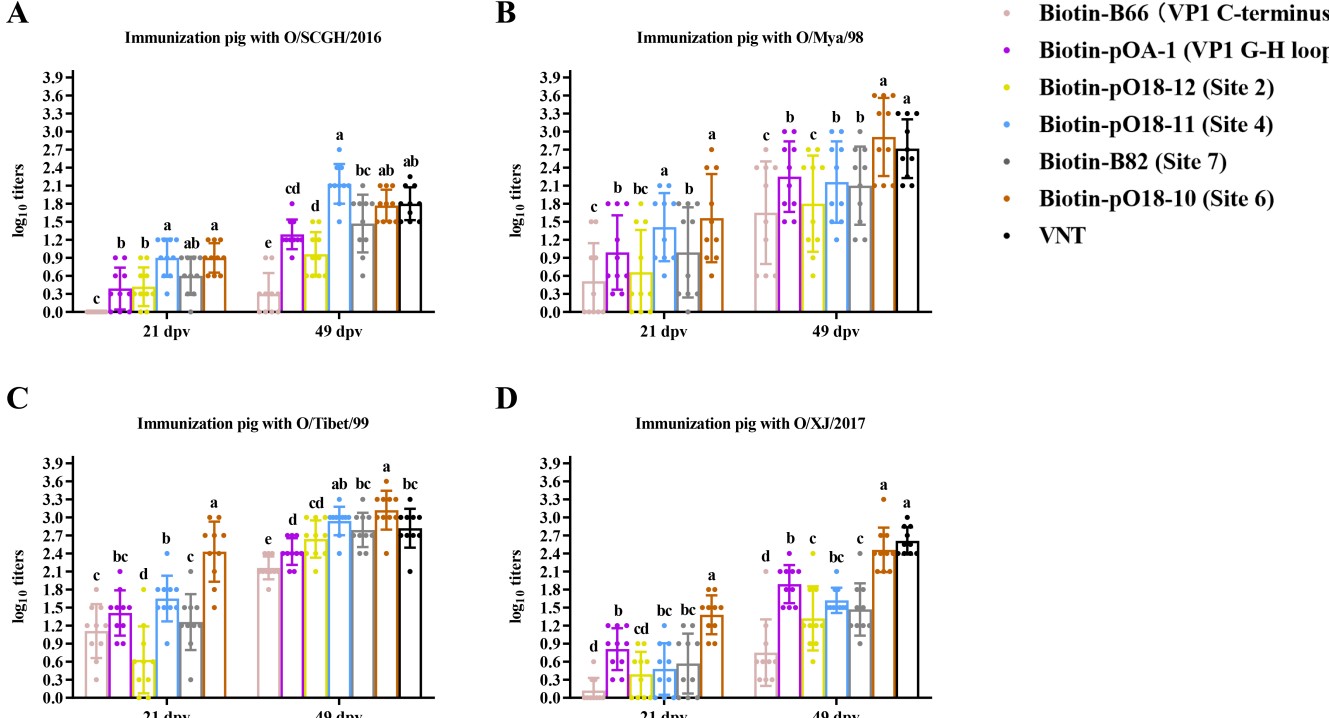

**FIG 4** Comparison of antibody abundance against various neutralizing sites after immunization of pigs with four lineages of FMDV serotype O vaccines. The same cELISA was used to detect pig serum samples collected at 21 and 49 dpv after immunization with the O/SCGH/2016 (A), O/Mya/98 (B), O/Tibet/99 (C), and O/XJ/2017 (D) vaccines. Antibody abundance was assessed using a site-directed cELISA targeting six epitopes described in Figure 2. Total NAb titers to O/Mya/98 virus at 49 dpv were determined through a VNT test. A one-way ANOVA (Tukey's multiple comparisons test) was the same as those described in Figure 2.

influence of the antigen used in cELISA, indicating these sites were evenly immunodominant sites for O/Mya/98, O/Tibet/99, and O/XJ/2017 in cattle.

Sheep displayed differential antibody responses to specific antigenic sites with variation in viral antigens after primary or booster vaccination. Sheep vaccinated with O/SCGH/2016 at 21 dpv showed the highest antibody responses to VP1 G-H loop without significant difference for that to site 4 ($P > 0.05$). Minimal antibody production was noted against the VP1 C-terminus (Fig. 3A). Sheep vaccinated with O/SCGH/2016 at 49 dpv exhibited the highest antibody abundance to site 4 (Fig. 3A; Fig. S2E). For those vaccinated with O/Mya/98 at 21 dpv, no significant differences were noted in antibody levels against the VP1 G-H loop, site 2, and site 4 ($P > 0.05$), while minimal antibody production was recorded against the VP1 C-terminus (Fig. 3B). By 49 dpv, sheep vaccinated with O/Mya/98 displayed significantly higher antibody responses to site 4 when tested with the homologous antigen, indicating its immunodominance (Fig. 3B). When using O/Tibet/99 antigen for cELISA, antibody abundance against site 4 showed no significant difference compared to the VP1 G-H loop and site 2 ($P > 0.05$) (Fig. S2F). Sheep vaccinated with O/Tibet/99 (21 dpv) and O/XJ/2017 (21 and 49 dpv) developed the highest antibody abundance against the VP1 G-H loop, which was significantly different from those observed in VP1 C-terminus, site 2, and site 4 ($P < 0.05$), confirming the immunodominance of the VP1 G-H loop for these strains (Fig. 3C and D; Fig. S2G and H). In sheep vaccinated with O/Tibet/99 at 49 dpv, both VP1 G-H loop and site 4 are immunodominant on provoking the antibody production ($P > 0.05$) (Fig. 3C; Fig. S2G).

Pigs also showed less efficiency in antibody production to different antigenic sites after primary vaccination. Variability in antibody responses to different sites was observed across virus lineages. Pigs vaccinated with O/SCGH/2016 at 21 and 49 dpv showed the highest antibody abundance to site 4, which was a significant difference from those to VP1 C-terminus, VP1 G-H loop, and site 2 ($P < 0.05$), indicating site 4 as an

immunodominant site (Fig. 4A; Fig. S2I). No antibody production was identified against the VP1 C-terminus (Fig. 4A). For pigs vaccinated with O/Mya/98 at 21 dpv, site 4 again showed the highest antibody abundance, significantly differing from responses to the VP1 C-terminus, VP1 G-H loop, and site 2 ($P < 0.05$), underscoring its immunodominance. At 49 dpv, pigs vaccinated with O/Mya/98 demonstrated elevated antibody levels against the VP1 G-H loop, site 2, and site 4 (Fig. 4B; Fig. S2J). Pigs vaccinated with O/Tibet/99 at 21 dpv showed the highest antibody levels for site 4, with the lowest for site 2; however, antibodies to all studied sites significantly increased at 49 dpv, with the highest responses observed for site 2 and site 4 (Fig. 4C; Fig. S2K). Lastly, pigs vaccinated with O/XJ/2017 at 21 and 49 dpv exhibited the highest antibody responses to VP1 G-H loop (Fig. 4D; Fig. S2L), which was similar to that observed in sheep.

## Immunogenicity of four lineages of FMDV serotype O strains in cattle, sheep, and pigs

Comparison of the antibody titers detected by cELISA for site 6 (probed by Biotin-pO18-10), LPBE, and VNT, the immunogenicity of four lineages of serotype O virus strains is summarized in Fig. 5. For each test, the differences in antibody induction of four virus strains were analyzed by one-way ANOVA (Tukey's multiple comparisons

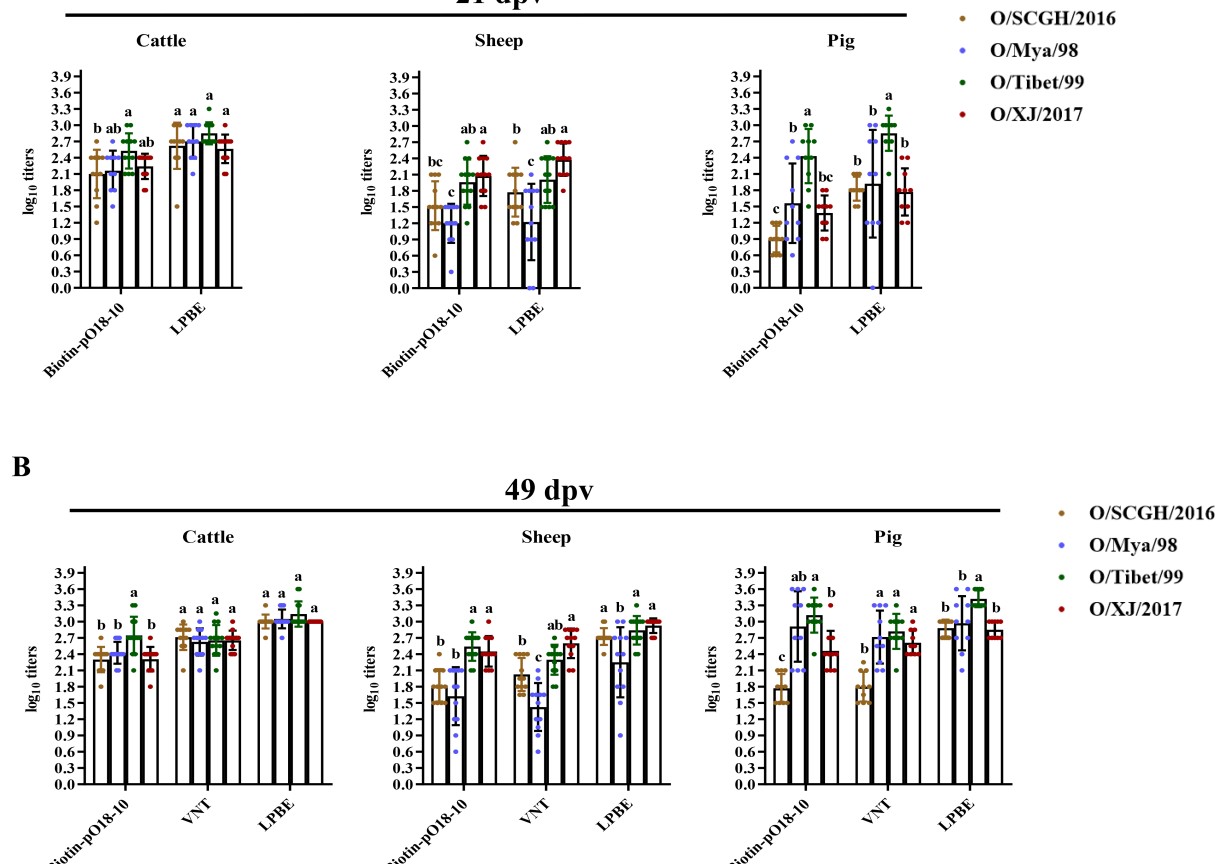

FIG 5 Immunogenicity analysis of four lineages of FMDV serotype O vaccines in different hosts. Serum samples were collected from cattle, sheep, and pigs after being vaccinated with four lineages of FMDV serotype O vaccines: O/Mya/98 (SEA topotype), O/SCGH/2016 (Cathay topotype), O/Tibet/99, and O/XJ/2017 (ME-SA topotype). A one-way ANOVA (Tukey's multiple comparisons test) was conducted to assess the differences in antibody titers between different virus lineages detected by Biotin-pO18-10-directed cELISA (E32 captured O/Mya/98) and LPBE at 21 dpv after vaccination of cattle, sheep, and pigs (A). The same analysis was conducted to assess the differences in antibody titers between different virus lineages detected by Biotin-pO18-10-directed cELISA (E32 captured O/Mya/98), VNT results (O/Mya/98), and LPBE at 49 dpv after vaccination of cattle, sheep, and pigs (B). Error bars represent 95% confidence intervals. In each figure, different letters indicate significant differences between the lineages during pairwise comparisons of the four topotype strains ($P < 0.05$).

test) to determine the statistical significance. Overall, the antibody titers detected by LPBE were higher than those obtained from VNT; however, LPBE did not effectively reflect the cross-neutralization effects among different viral strains, particularly in sheep and pigs (Fig. 5A and B). The average NAb titers detected by Biotin-pO18-10 (for site 6) were consistent with those detected by LPBE and VNT, indicating a potential alternative method for evaluation of the cross-protection among strains. Generally, all four strains induced robust immune responses in vaccinated cattle after both primary and booster vaccination, except that slightly higher antibody titers were detected in O/Tibet/99-vaccinated cattle by Biotin-pO18-10 cELISA. In sheep, both O/XJ/2017 and O/Tibet/99 exhibited enhanced antibody responses compared to O/Mya/98. O/Tibet/99 elicited a strong antibody response in pigs even at 21 dpv, whereas O/Mya/98 and O/XJ/2017 required booster immunizations to reach antibody levels comparable to those elicited by O/Tibet/99, indicating superior immunogenicity of O/Tibet/99 in pigs. Overall, the O/SCGH/2016 strain demonstrated inferior immunogenicity in pigs compared to the other three strains.

## Correlation analysis of the two outcomes between each of six site-directed cELISAs and VNT

A total of 139 serum samples were collected from cattle, sheep, and pigs ($n$ = 139) at 49 dpv after booster vaccination with four lineages of different serotype O virus antigens. Site-directed cELISAs were performed by capturing O/Mya/98 and O/Tibet/99 virus antigens, respectively, to detect the antibody abundance of each antigenic site. Serum samples were also detected by VNT to neutralize O/Mya/98 and O/Tibet/99

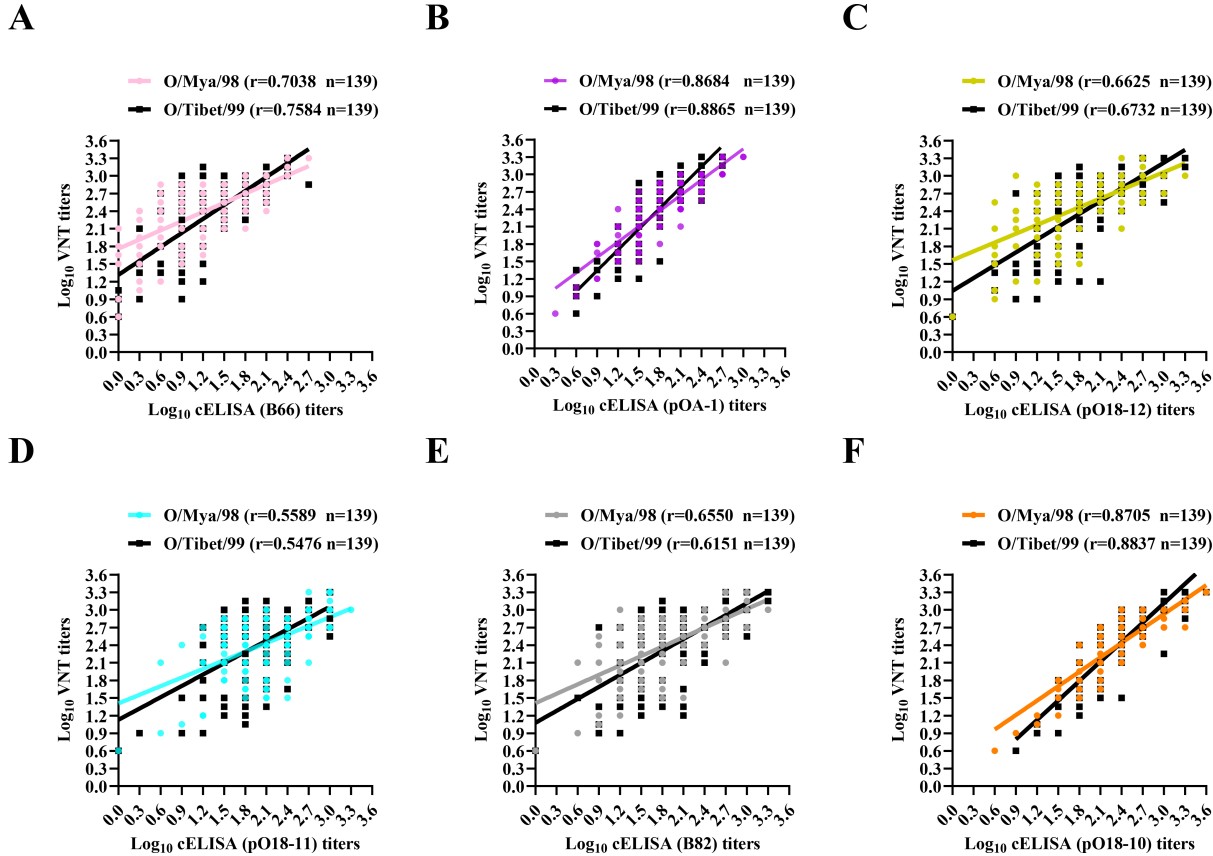

**FIG 6** Correlation of the titers obtained using the VNT and cELISA. Correlation between VNT titers and cELISA titers probed by Biotin-B66 (A), Biotin-pOA-1 (B), Biotin-pO18-12 (C), Biotin-pO18-11 (D), Biotin-B82 (E), and Biotin-pO18-10 (F) using two antigens, respectively. Pearson's correlation coefficient was calculated using GraphPad Prism (version 9.5.1) software (San Diego, CA, USA). n, number of serum samples tested; r, correlation coefficient. Solid lines represent the fitted curve. Note: some dots represent more than one serum sample.

viruses separately. The cELISAs and VNT titers were relatively correlated when detected, respectively, with O/Mya/98 and O/Tibet/99 antigens with Pearson correlation coefficients calculated for each comparison (Fig. 6A through F). Notably, antibody titers detected by cELISA targeted to VP1 G-H loop (by Biotin-pOA-1) and site 6 (by Biotin-pO18-10) displayed particularly higher correlation coefficients (r = 0.8684/0.8865, 0.8705/0.8837) with VNT titers than the other sites (r = 0.5476~0.7584). The mean values of cELISAs and VNT titers for two antigens were as follows: Biotin-pOA-1 (1.84/1.76), Biotin-pO18-10 (2.37/2.32), and VNT (2.41/2.37). Obviously, cELISA titers detected by Biotin-pO18-10 were close to and well correlated to those by VNT, indicating its potential as an alternative method for assessing protective immunity in animals inoculated with FMD inactivated vaccines.

## DISCUSSION

In this study, we selected 35 bovine- or porcine-derived NAbs that showed good reactivity with FMDV serotype O antigens to develop site-directed cELISAs to study the spatial distribution of different antigenic sites. Twenty-seven NAbs, which showed self-competition in cELISA based on whole-virus antigens, were selected to carry out pairwise cELISA to reclassify the antigenic sites according to their spatial hindrance effects. Our findings revealed that four antigenic sites were spatially separated, which were VP1 C-terminus, VP1 G-H loop, conventional antigenic site 2 (in VP2), and site 4 (in VP3). Additionally, we identified two new antigenic sites characterized by NAbs with broad contacts on the capsids, which were site 6 involving multiple binding residues in VP1 and VP3, and site 7 involving multiple binding residues in VP2 and VP3. For the conventional antigenic site 3 in VP1, suitable NAbs for pairwise cELISA were not available, leaving the spatial relationship of site 3 with other antigenic sites undetermined. The results suggest that antigenic sites are not exclusively localized to individual VP1, VP2, and VP3 proteins, with transitional antigenic sites, such as site 6 (VP1 and VP3) and site 7 (VP2 and VP3), being prevalent and spanning across two proteins. While transitional antigenic sites connecting VP1 and VP2 were not identified in this investigation, their potential existence cannot be ruled out. The location of site 6 on two adjacent pentamers of FMDV shows that the G-H loop in VP1 adopts an elevated conformation, which is separated from VP1 C-terminus (Fig. 1B). Notably, previous research suggests significant conformational flexibility and structural disorder in the VP1 G-H loop (37–39), along with flexibility in the VP1 C-terminus region, allowing movement in various orientations. To ascertain whether the epitopes recognized by NAbs to site 6 are confined to a single protomer or span multiple protomers, further investigations employing advanced techniques such as cryo-electron microscopy are essential in the future.

Steric interference was inferred from the substantial size of antibody molecules and the proximity of antigenic sites on the FMDV surface, which seemed to be measurable by inhibition rates in cELISA using different monoclonal antibodies. Previous studies employing cELISA had demonstrated that antibody C2 targeting antigenic site 4 significantly competed with antibody C48 targeting antigenic site 2, despite binding to different antigenic sites indicating the spatial adjacency of these two sites (20). In this study, we classified 27 NAbs into six classes based on their steric interference with each other when binding to antigenic sites on two distinct antigens of serotype O viruses (ME-SA and SEA topotype) (Fig. 1A; Fig. S1). Although competition rates varied for certain antibodies when tested with two different antigens, the consistent results were observed in cELISA, which likely reflected the slight differences in antigenic structures. For instance, competition rates for NAbs pOA-1, pO18-40, and pOA-7 against Biotin-pOA-13 were recorded at 100%~101% on O/Mya/98 antigen, whereas it ranged from 71% to 77% on O/Tibet/99. Notably, NAbs binding to VP1 G-H loop (pOA-1, pO18-39, pO18-40, pOA-7, pOA-20, and pOA-13) exhibited 58%~81% competition rates with Biotin-B57 on O/Mya/98 antigen, compared to approximately 21%–51% on O/Tibet/99 antigen. These results indicate that VP1 G-H loop is positionally closer to the binding site of NAb B57 (site 7) in O/Mya/98 compared with O/Tibet/99 antigen. Similarly, NAbs binding to VP1

G-H loop (pOA-1, pO18-39, pO18-40, and pOA-7) demonstrated 53%~69% competition against Biotin-B73 (binding to site 2) tested with both antigens, suggesting that the epitope recognized by B73 is structurally closer to VP1 G-H loop than other epitopes (pOA-2 and pO18-12) within site 2. Given the diversity of antibodies, competitive interactions among antibodies of the same class may be varied. For example, NAbs B57 and F28 binding to site 7 exhibited 30% and 44% competition with Biotin-pO18-16 in cELISA tested with O/Mya/98 antigen, respectively, although these three antibodies were structurally attributed to the same class; similar results were also observed for NAbs B73 and Biotin-pO18-12, both binding site 2 with only a 27% competition rate in cELISA (Fig. 1A). This variability suggests that the affinity of antibodies in the same classes may potentially impact their competitive dynamics in cELISA and also influence their neutralizing efficacy against the virus. From the results of pairwise cELISAs, we can observe that antigenic site 2, site 4, and site 7 are spatially independent of antigenic site 1, site 5, or even site 6 (related to integrin-binding site) designated with classes of NAbs in this study, which may imply a different virus-neutralizing mechanism for those NAbs targeting to site 2, site 4, and site 7, other than blocking virus attachment of cellular receptor. Following immunization, animal sera contain a diverse array of antibodies targeting various epitopes. During competitive reactions with serum samples using six representative antibodies, we propose that preferential competition occurs among antibodies binding to the same epitope or spatially adjacent epitopes, which can be indicated by the inhibition rate detected by cELISA.

After immunizing three hosts with four strains of serotype O virus antigens (O/SCGH/2016, O/Mya/98, O/Tibet/99, and O/XJ/2017), differences in antibody responses targeting six antigenic sites in the sera were observed, indicating potential variations in the immunodominant antigenic sites among different virus strains. The O/Mya/98 and O/Tibet/99 strains exhibited more balanced antibody responses to each site, with no obvious immunodominant sites. Site 4 emerged as the immunodominant epitope for the O/SCGH/2016 strain, suggesting significant conformational changes in VP3 protein and immune evasion features for this swine-adapted virus. In contrast, the VP1 G-H loop served as the immunodominant epitope for the O/XJ/2017 strain, which is the receptor-binding domain (RBD) of FMDV and the primary target for potent NAbs to block viral attachment. RGD motif in the VP1 G-H loop is essential for integrin receptor recognition and usually recognized as an immunodominant neutralizing site (29–31). However, vaccination with the VP1 G-H loop peptide alone did not provide consistent and complete protection in cattle without the association with the VP1 C-terminus, indicating a synergistic effect between these two fragments (40–42). Competitive ELISA based on monoclonal antibodies indicates that pigs exhibit lower antibody responses to VP1 C-terminus compared to ruminants (32). This phenomenon was also confirmed in this study in sheep and pigs with the exception of the O/Tibet/99 virus strain, indicating its good immunogenicity. Thus, our study reveals the different levels of antibody responses to six antigenic sites and their variation among four lineage strains of serotype O viruses in different animal species, which will provide valuable information for designing more potent vaccines against FMD. The illustration of the spatial relationship of different antigenic sites also makes a good basis for developing an alternative test to accurately detect neutralizing antibodies. Competitive ELISA detection of antibodies to site 6 showed good correlation with VNT results, indicating a comprehensive effect by NAb pO18-10 covering VP1 and VP3 regions on capsids to reflect the total neutralizing antibody level after vaccination, which needs to be further evaluated with more serum samples.

Studies had shown that although certain high-quality commercial FMD vaccines elicited robust immune protection in cattle, they were not always effective at inducing adequate immune responses in pigs, suggesting significant differences in the immune mechanisms to FMDV between these two species (43–45). Specifically, cattle typically exhibited a more pronounced immune memory response following FMDV infection, and their immune systems tended to generate long-lasting neutralizing antibodies, thus

providing rapid and sustained protection against reinfection (46, 47). In contrast, pigs exhibited comparatively weaker immune memory responses, with slower kinetics and reduced persistence of neutralizing antibody production (48). Continual variation in FMDV antigenic sites could lead to a poor match between vaccine strains and circulating field strains. This phenomenon resembled the antigenic drift observed in the hemagglutinin (HA) protein of influenza A viruses, highlighting the need for frequent updates to vaccine formulations to ensure sustained protection (49, 50). This study found that the immunodominance patterns targeting different sites of the same FMDV strain had varied significantly after immunization in different hosts. This observation aligned with previous studies in influenza virus immunization, where C57BL/6 and BALB/c mice displayed marked differences in immunodominance hierarchies (51, 52). Such species-specific immunodominance emphasized the complexity of host–pathogen interactions and underscored the challenges of developing broadly effective vaccines.

In summary, this study clarified the spatial relationship of six neutralizing antigenic sites on the virus capsid of FMDV serotype O. NAbs play a crucial role as the primary protective component against FMDV infection. Systematically comparing epitope immunodominance in sera from cattle, sheep, and pigs immunized with different lineages of serotype O is essential for designing vaccine antigens with broad antigenic coverage. This comparison aids in selecting optimal vaccine strains and developing more effective methods for the detection of NAbs to evaluate the vaccine protection against FMD.

## MATERIALS AND METHODS

### Serum samples

The serum samples used in this study were previously collected from animal experiments conducted by the management guidelines of the Gansu Provincial Ethics Review Committee (License No. SYXK-89 GAN-2014-003) (53). All animals used in the present study were humanely bled. Vaccines used for animal inoculation were prepared by purification and accurate quantification of 146S antigen of each strain of FMDV to contain 6 µg antigen per dose in 2 mL vaccine. A total of 279 serum samples were collected from vaccinated animals as follows: 50 10-month-old cattle, 50 12-month-old sheep, and 40 2-month-old white pigs were randomly divided into four groups. Each group received a 2 mL dose of O/SCGH/2016 (GenBank No. KX161429), O/Mya/98 (GenBank No. JN998085), O/Tibet/99 (GenBank No. AJ539138), or O/XJ/2017 (GenBank No. MF461724) vaccine on days 0 and 21. Serum samples were collected at 21 and 49 dpv. After the second immunization, one sheep identified as 2188#, immunized with the O/XJ/2017 vaccine, died of diarrhea on 37 dpv, resulting in a total of 99 serum samples from 50 sheep being used in this study.

### Cloning, expression, and purification of monoclonal antibodies

Bovine-derived non-neutralizing antibody E32 and 12 NAbs A19, B57, B66, B73, B74, B77, B82, C4, C5, E18, F28, and F169 were previously generated in our laboratory by single B-cell antibody techniques (35). A total of 23 porcine-derived NAbs pOA-1, pOA-2, pOA-6, pOA-7, pOA-13, pOA-20, pO18-2, pO18-8, pO18-10, pO18-11, pO18-12, pO18-16, pO18-17, pO18-20, pO18-24, pO18-26, pO18-28, pO18-39, pO18-40, pO18-52, pO18-53, pO18-54, and pO18-57 were screened from a single B-cell antibody library (36). Monoclonal antibodies were expressed and purified according to previously described methods (35, 36). Purified antibodies were analyzed by reducing 12% sodium dodecyl sulfate-polyacrylamide gel electrophoresis (SDS-PAGE), and proteins were detected using Coomassie Brilliant Blue staining. The concentration of expressed antibodies was determined by measuring the absorbance values at a wavelength of 280 nm (A280).

## Biotinylation of NAbs

Biotinylation of 35 NAbs used in this study was performed with EZ-Link Sulfo-NHS-LC-biotin reagent (Thermo Fisher Scientific, USA), following the manufacturer's instructions. After labeling, the antibodies were dialyzed three times against phosphate-buffered saline (PBS), and their concentrations were measured before use.

## Development of cELISA

The optimal concentrations of the capture antibody (E32), 146S antigens (O/Mya/98 and O/Tibet/99), Biotin-NAbs, and horseradish peroxidase (HRP)-streptavidin were determined using the checkerboard method for cELISA. Initially, E32 was diluted in carbonate-bicarbonate buffer (pH 9.6) to a final concentration of 0.5 µg/mL and coated onto 96-well ELISA plates overnight at 4℃. After three washes with PBST (PBS with 0.05% Tween-20), the 146S antigens at 1 µg/mL were added into the plates and captured for 2 h at room temperature. After three washes with PBST, the plates were blocked with PBS buffer containing 5% sucrose and 1% bovine serum albumin (BSA) at 37℃ for 1 h. After three washes, neutralizing antibodies or serum samples were diluted in 50 µL of PBS (ranging from 1:4 to 1:512) and mixed with an equal volume of biotinylated antibody diluted in PBS (final dilution ranging from 1:8 to 1:1,024), followed by incubation at 37℃ for 1 h. After five washes with PBST, HRP-conjugated streptavidin (diluted 1:30,000) was added, and the plates were incubated at 37℃ for 15 min. Following another five washes with PBST, 100 µL of tetramethylbenzidine (TMB) substrate was added to each well, and the plates were incubated at 37℃ for 15 min. The color reaction was stopped by adding 100 µL of 2 M $H_2SO_4$, and the absorbance at 450 nm (A450) was measured using an automatic microplate reader (BioTek).

To identify the optimal concentration of the 35 Biotin-NAbs for cELISA, the antibodies were first diluted in PBS to 5 µg/mL and then subjected to 16 subsequent twofold dilutions. Each well was filled with 100 µL of the diluted Biotin-NAb and was incubated at 37℃ for 1 h. The optimal concentration for cELISA was determined to be the dilution that produced an A450 value of approximately 2.0.

The competitive interactions among 27 NAbs were assessed using pairwise cELISAs. Competitor NAbs were initially diluted to 50 µg/mL and then subjected to eight rounds of twofold dilutions to compete with Biotin-NAbs. Each ELISA plate included four wells of PBS control (100% nonreactivity) and four wells of Biotin-NAb control for each pairwise cELISA. The highest and lowest absorbance values at A450 were excluded, and the average of the remaining two wells was calculated. The percentage inhibition of competitor NAb to Biotin-NAb binding was calculated using the formula: Inhibition (%) = (A450 of Biotin-NAb control – A450 of competitor NAb) / (A450 of Biotin-NAb control – A450 of PBS control). The inhibition rates of all competitor antibodies at the same optimal concentration were plotted for further analysis.

For measuring antibody titers in cattle, sheep, and pig sera, each ELISA plate included a positive-control serum of known titer, tested in duplicate with twofold dilutions ranging from 1:4 to 1:512. A negative-control serum (from unvaccinated naive animals) was tested in four duplicate twofold dilutions from 1:4 to 1:32. Four wells served as antigen controls (100% reactivity), while two wells were used as reaction blanks without 146S antigen and serum. Antibody titers were calculated as the reciprocal (log10) of the serum dilution that resulted in 50% of the absorbance recorded in the antigen control wells. For the assay to be valid, the mean A450 value of the antigen controls should be 2.0 ± 0.5, the titer (log10) of the strong positive-control serum should be 2.85 ± 0.3, and the titer (log10) of the negative-control serum should be less than 0.9.

## Liquidphase blocking ELISA

FMDV serotype O antibodies in serum samples were detected using a commercial LPBE kit from Lanzhou Veterinary Research Institute of the Chinese Academy of Agricultural Sciences. This kit employed 146S antigen along with rabbit and guinea pig polyclonal

antisera prepared from the FMDV O/Mya/98 strain. All procedures were conducted according to the manufacturer's instructions (54). For valid experimental conditions, the positive-control serum should exhibit a titer (log10) of $3 \pm 0.3$, while the negative-control serum should have a titer (log10) of less than 0.9.

## Virus-neutralizing test

Serum samples from animals immunized with four representative O strains (O/SCGH/2016, O/Mya/98, O/Tibet/99, and O/XJ/2017) at 49 dpv were detected by VNT on BHK-21 monolayer cells as previously described (55). The strains used in the VNT were the same as those used for coating in the ELISA for serum detection. Briefly, serum samples were serially diluted twofold in a 96-well plate and were incubated with 100 tissue culture infective doses ($TCID_{50}$) of the virus at 37°C for 1 h. Subsequently, $5 \times 10^4$ BHK-21 cells per well were added and incubated at 37°C with 5% $CO_2$ for 72 h. Each plate included wells with virus controls at 0.1, 1, 10, and 100 $TCID_{50}$, as well as wells with normal cells. Neutralization potency was calculated as the reciprocal of the highest serum dilution that neutralized 100 $TCID_{50}$ of FMDV in 50% of the wells. Serum titers $\geq 1.65$ were considered positive, while those $\leq 0.9$ were considered negative controls.

## Statistical analysis

GraphPad Prism software version 9.5.1 was used for the statistical analysis of the data. To compare the results presented in the bar chart, a one-way ANOVA (Tukey's multiple comparisons test) was conducted to assess the statistical significance of the differences. The letter-marking method was employed to identify significant variations. Within each group, different lowercase letters indicate statistically significant differences between pairwise samples ($P < 0.05$), while the same lowercase letter signifies no significant difference between pairwise samples ($P > 0.05$). Pearson's correlation coefficient was used to assess the relationship between cELISA titers and VNT titers, with $P < 0.05$ considered statistically significant.

## ACKNOWLEDGMENTS

This work was supported by the National Key R & D Program of China (2021YFD1800300 to Z.L. and X.L.).

## AUTHOR AFFILIATIONS

[1]National Key Laboratory of Agricultural Microbiology, Hubei Hongshan Laboratory, Huazhong Agricultural University, Wuhan, Hubei, China
[2]Key Laboratory of Preventive Veterinary Medicine in Hubei Province, The Cooperative Innovation Center for Sustainable Pig Production, Wuhan, Hubei, China
[3]College of Veterinary Medicine, Huazhong Agricultural University, Wuhan, Hubei, China
[4]Hubei Jiangxia Laboratory, Wuhan, Hubei, China
[5]State Key Laboratory for Animal Disease Control and Prevention, College of Veterinary Medicine, Lanzhou University, National Foot-and-Mouth Diseases Reference Laboratory, Lanzhou Veterinary Research Institute, Chinese Academy of Agricultural Sciences, Lanzhou, China

## AUTHOR ORCIDs

Qiongqiong Zhao http://orcid.org/0009-0006-4934-4738
Xueqing Ma http://orcid.org/0009-0003-6780-0061
Yimei Cao http://orcid.org/0009-0000-1267-9741
Kun Li http://orcid.org/0000-0001-8307-634X
Xiangmin Li http://orcid.org/0000-0003-1412-5069
Zengjun Lu http://orcid.org/0000-0001-9138-692X

## FUNDING

| Funder | Grant(s) | Author(s) |
|---|---|---|
| National Key Research and Development Program of China | 2021YFD1800300 | Xiangmin Li |
| | | Zengjun Lu |

## AUTHOR CONTRIBUTIONS

Qiongqiong Zhao, Data curation, Formal analysis, Visualization, Writing – original draft, Investigation, Methodology, Supervision, Software | Fengjuan Li, Methodology | Shulun Huang, Software | Xiangchuan Xing, Data curation | Ying Sun, Data curation | Pinghua Li, Data curation | Huifang Bao, Data curation | Yuanfang Fu, Conceptualization | Pu Sun, Conceptualization | Xingwen Bai, Conceptualization | Hong Yuan, Conceptualization | Xueqing Ma, Conceptualization | Zhixun Zhao, Conceptualization | Jing Zhang, Conceptualization | Jian Wang, Conceptualization | Tao Wang, Conceptualization | Dong Li, Conceptualization | Qiang Zhang, Conceptualization | Ping Qian, Supervision | Yimei Cao, Conceptualization, Methodology, Supervision | Kun Li, Conceptualization, Methodology, Writing – review and editing | Xiangmin Li, Funding acquisition, Writing – review and editing | Zengjun Lu, Writing – review and editing

## DATA AVAILABILITY

The sequences of 13 bovine-derived antibodies A19, B57, B66, B73, B74, B77, B82, C4, C5, E18, E32, F28, and F169, as well as six porcine-derived antibodies pOA-1, pOA-2, pOA-6, pOA-7, pOA-13, and pOA-20 have been published previously (35, 36). The sequences of 17 porcine-derived antibodies pO18-2, pO18-8, pO18-10, pO18-11, pO18-12, pO18-16, pO18-17, pO18-20, pO18-24, pO18-26, pO18-28, pO18-39, pO18-40, pO18-52, pO18-53, pO18-54, and pO18-57 can be obtained by contacting the corresponding author if needed. All other data supporting the findings of this study are available from the corresponding authors upon request.

## ADDITIONAL FILES

The following material is available online.

### Supplemental Material

**Supplemental figures and tables (Spectrum03344-24-s0001.docx).** Fig. S1 to S3 and Tables S1 to S8.

### Open Peer Review

**PEER REVIEW HISTORY (review-history.pdf).** An accounting of the reviewer comments and feedback.

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
