## [Reviewer comments · Microbiology Spectrum]

Microbiology Spectrum

Remapping the spatial distribution of neutralizing sites and their immunodominance on the capsid of different topotypes of FMDV serotype O by site-directed competitive ELISA for detection of neutralizing antibodies

Qiongqiong Zhao, Fengjuan Li, Shulun Huang, Xiangchuan Xing, Ying Sun, Pinghua Li, Huifang Bao, Yuanfang Fu, Pu SUN, Xingwen BAI, Hong Yuan, Xueqing Ma, Zhixun Zhao, Jing Zhang, Jian Wang, Tao Wang, Dong Li, Qiang Zhang, Ping Qian, Yimei Cao, Kun Li, Xiangmin Li, and Zengjun Lu

Corresponding Author(s): Zengjun Lu, Lanzhou Veterinary Research Institute

Review Timeline:

Submission Date:	December 20, 2024
Editorial Decision:	January 21, 2025
Revision Received:	February 10, 2025
Accepted:	March 3, 2025

Editor: Gabriel Parra

Reviewer(s): Disclosure of reviewer identity is with reference to reviewer comments included in decision letter(s). The following individuals involved in review of your submission have agreed to reveal their identity: Kelsey A Pilewski (Reviewer #1); Shawn Babiuk (Reviewer #2)

Transaction Report:

DOI: <https://doi.org/10.1128/spectrum.03344-24>

Re: Spectrum03344-24 (Remapping the spatial distribution of neutralizing sites and their immunodominance on the capsid of different topotypes of FMDV serotype O by newly developed site-directed competitive ELISA for detection of neutralizing antibodies.)

Dear Dr. Zengjun Lu:

Thank you for submitting your manuscript to Spectrum. We have received evaluations from two experts in the field, both of whom recognize the novelty and potential impact of your work. However, they suggest several modifications that could help strengthen the study further. I am also providing some additional recommendations for your consideration.

Revision Guidelines

Sincerely,
Gabriel Parra
Editor
Microbiology Spectrum

Editor's Comments:

1. Please consider removing the term "antigenic" when used in conjunction with "epitopes" (e.g., lines 104, 152, 336). Since epitopes inherently elicit antibodies, the term "antigenic" is redundant.
2. Please rephrase "neutralizing epitopes" (line 118). It would be more accurate to state that antibodies targeting these epitopes are responsible for neutralizing the virus.

3. The concept of distinct strain-specific (and potentially species-specific) immunodominance is intriguing and seems to differ from what has been described for influenza viruses (e.g., *Nat Immunol.* 2017 Apr;18(4):456-463). We encourage you to include a brief discussion of your findings in the context of other viral pathogens.

4. Figure 1: For a more comprehensive interpretation of your data, I suggest rendering the spatial distribution of the antigenic sites within the context of the entire virion.

Reviewer #1 (Comments for the Author):

This manuscript by Zhao et al. describes antibody responses to neutralizing sites on the capsid of an important animal pathogen, FMDV. The authors describe the spatial relationship between monoclonal antibodies recognizing distinct sites on the FMDV capsid and the relative abundance of these antibodies in vaccinated animals. Using competition ELISA, the authors estimate the immunodominance of these capsid sites in polyclonal sera and compare topotype immunogenicity in 3 natural host animals. This manuscript is generally well-written and addresses gaps in knowledge for an important animal pathogen. However, there are several points that require clarity before publication of this manuscript.

Specific comments:

Please remove "newly developed" from the title, as this assay has been described before.

Description of how Figure 1B was generated should be discussed or revised. It appears to be the important residues for each site modeled on the structure of the FMDV capsid. If so, the figure legend and description in the text (lines 211, 332) should be revised to reflect that this is modelling, not simulation. Please provide the PDB ID for the structure file used to create the figure. Additionally, it would help interpretation to include the color legend for VP1-3 in the figure.

Figures 2-4: As described, it seems that lower titers in the presence of biotinylated antibody would indicate more competition and thus a higher proportion of antibodies recognizing that site are present in the polyclonal serum. My interpretation of the low titers in the presence of B66 suggests that the C-terminus of VP1 is immunodominant. Could the authors please clarify this?

Please revise Figures 2-5 to use the standard system for statistics reporting (* for $p < 0.05$, ** for $p < 0.01$, *** for $p < 0.001$, **** for $p < 0.0001$). Please indicate which columns are being compared for each statistic.

It would be helpful to include the site specificities beside the representative biotin mAbs in figures 2-4 to ease reader interpretation.

Please provide a color legend in figure 6.

The data for distinct strains should be separated or labeled in figure 6, as you've just demonstrated that site-specific responses differ across these topotypes. For example, color the O/Mya/98 and O/Tibet/99 data points differently or calculate correlation for each strain.

Reviewer #2 (Comments for the Author):

The author's present work characterizing the neutralizing epitopes from different FMDV O topotypes.

Minor issues

Line 271 Change "None" to "No"

Line 303 Pearson

Line 332 Change "aroused" to "inferred"

Line 336 Change "despite they bind to different antigenic sites" to "despite binding to different antigenic sites"

Line 395 The differences in NAb production in cattle among four strains of serotype O viruses were less than that in sheep and pigs indicating the high efficiency of humoral immune responses in cattle.

I do not believe that the efficiency of humoral immune responses in cattle are really different compared to sheep and pigs. For some reason with FMD pigs are more difficult to induce protective immunity compared to cattle. The reason for this is not really known. Discuss this and add a reference.

Line 397-401 These lines do not really add anything in the discussion and could be removed.

This manuscript by Zhao et al. describes antibody responses to neutralizing sites on the capsid of an important animal pathogen, FMDV. The authors describe the spatial relationship between monoclonal antibodies recognizing distinct sites on the FMDV capsid and the relative abundance of these antibodies in vaccinated animals. Using competition ELISA, the authors estimate the immunodominance of these capsid sites in polyclonal sera and compare topotype immunogenicity in 3 natural host animals. This manuscript is generally well-written and addresses gaps in knowledge for an important animal pathogen. However, there are several points that require clarity before publication of this manuscript.

Specific comments:

Please remove “newly developed” from the title, as this assay has been described before.

Description of how Figure 1B was generated should be discussed or revised. It appears to be the important residues for each site modeled on the structure of the FMDV capsid. If so, the figure legend and description in the text (lines 211, 332) should be revised to reflect that this is modelling, not simulation. Please provide the PDB ID for the structure file used to create the figure. Additionally, it would help interpretation to include the color legend for VP1-3 in the figure.

Figures 2-4: As described, it seems that lower titers in the presence of biotinylated antibody would indicate more competition and thus a higher proportion of antibodies recognizing that site are present in the polyclonal serum. My interpretation of the low titers in the presence of B66 suggests that the C-terminus of VP1 is immunodominant. Could the authors please clarify this?

Please revise Figures 2-5 to use the standard system for statistics reporting (* for $p < 0.05$, ** for $p < 0.01$, *** for $p < 0.001$, **** for $p < 0.0001$). Please indicate which columns are being compared for each statistic.

It would be helpful to include the site specificities beside the representative biotin mAbs in figures 2-4 to ease reader interpretation.

Please provide a color legend in figure 6.

The data for distinct strains should be separated or labeled in figure 6, as you've just demonstrated that site-specific responses differ across these topotypes. For example, color the O/Mya/98 and O/Tibet/99 data points differently.

Responds to the reviewers' comments

Dear Editor,

On behalf of my co-authors, I would like to express our great appreciation to you and reviewers. Thank you very much for your reading and comments of our manuscript entitled “Remapping the spatial distribution of neutralizing sites and their immunodominance on the capsid of different topotypes of FMDV serotype O by newly developed site-directed competitive ELISA for detection of neutralizing antibodies” (Manuscript ID: Spectrum03344-24). We have revised our paper carefully according to the reviewers’ comments and responded to all questions point-to-point. We hope that the correction will meet with approval. Please contact me if you have any questions on this paper. Below is our response to reviewer’s helpful and constructive comments, all changes have been highlighted in red in the revised manuscript:

Responds to the reviewers' comments:

Editor's Comments:

1. Please consider removing the term "antigenic" when used in conjunction with "epitopes" (e.g., lines 104, 152, 336). Since epitopes inherently elicit antibodies, the term "antigenic" is redundant.

Response: Thanks for your suggestion. We changed it following your advice. See line 100, 148, 331.

2. Please rephrase "neutralizing epitopes" (line 118). It would be more accurate to state that antibodies targeting these epitopes are responsible for neutralizing the virus.

Response: Thanks for your suggestion. We corrected the sentence as follows: “This region is also a highly variable structural domain containing multiple epitopes responsible for virus neutralization, such as antigenic sites 1 and 5.” See line 113-115.

3. The concept of distinct strain-specific (and potentially species-specific) immunodominance is intriguing and seems to differ from what has been described for influenza viruses (e.g., Nat Immunol. 2017 Apr;18(4):456-463). We encourage you to include a brief discussion of your findings in the context of other viral pathogens.

Response: Thanks for your suggestion. We added discussion that “Continual variation in FMDV antigenic sites could lead to a poor match between vaccine strains and circulating field strains. This phenomenon resembled antigenic drift observed in the hemagglutinin (HA) protein of influenza A viruses, highlighting the need for frequent updates to vaccine formulations to ensure sustained protection (Hensley SE. *Science*. 2009; Altman MO. *Viral Immunol*. 2018). This study found that the immunodominance patterns targeting different sites of the same FMDV strain had varied significantly after immunization in different hosts. This observation aligned with previous studies in influenza virus immunization, where C57BL/6 and BALB/c mice displayed marked differences in immunodominance hierarchies (Angeletti D. *Nat Immunol*. 2017; Liu STH. *J Clin Invest*. 2018). Such species-specific immunodominance emphasized the complexity of host-pathogen interactions and underscored the challenges of developing broadly effective vaccines.” See line 398-406.

4. Figure 1: For a more comprehensive interpretation of your data, I suggest rendering the spatial distribution of the antigenic sites within the context of the entire virion.

Response: Thanks for your suggestion. The entire FDMV virion has 60 protomers, we cannot model all antigenic sites on the entire virion, this is not so easy to do. So, we show the antigenic sites on the two adjacent protomers.

To improve the accuracy of the descriptions, we have made some minor adjustments to the language, which are highlighted in the "Marked-Up Manuscript" file.

Reviewer #1 (Comments for the Author):

This manuscript by Zhao et al. describes antibody responses to neutralizing sites on the capsid of an important animal pathogen, FMDV. The authors describe the spatial relationship between monoclonal antibodies recognizing distinct sites on the FMDV capsid and the relative abundance of these antibodies in vaccinated animals. Using competition ELISA, the authors estimate the immunodominance of these capsid sites in polyclonal sera and compare topotype immunogenicity in 3 natural host animals. This manuscript is generally well-written and addresses gaps in knowledge for an important animal pathogen. However, there are several points that require clarity before publication of this manuscript.

Specific comments:

Please remove "newly developed" from the title, as this assay has been described before.

Response: Thanks for your suggestion. We changed it following your advice. See line 2.

Description of how Figure 1B was generated should be discussed or revised. It appears to be the important residues for each site modeled on the structure of the FMDV capsid. If so, the figure legend and description in the text (lines 211, 332) should be revised to reflect that this is modelling, not simulation. Please provide the PDB ID for the structure file used to create the figure. Additionally, it would help interpretation to include the color legend for VP1-3 in the figure.

Response: Thank you for your suggestion. We have implemented the requested changes. Specifically, we have included the PDB ID used to generate the structural model for Figure 1B, which is PDB:1FOD, in the figure legend (lines 684-685). Furthermore, we have revised both the figure legend and the corresponding text descriptions for Figure 1B (lines 683-686, 206-208, 327-329), and we have added a color legend for VP1-3 within the figure.

Figures 2-4: As described, it seems that lower titers in the presence of biotinylated antibody would indicate more competition and thus a higher proportion of antibodies recognizing that site are present in the polyclonal serum. My interpretation of the low titers in the presence of B66 suggests that the C-terminus of VP1 is immunodominant. Could the authors please clarify this?

Response: Thanks. In these site-directed cELISA, lower titers suggest a lower proportion of antibodies in polyclonal serum targeted to this specific site, while higher titers indicate a larger proportion of antibodies in serum samples targeted to the sites. A low titer in serum samples probed by biotin-B66 implies that the C-terminus of VP1 is not immunodominant.

Please revise Figures 2-5 to use the standard system for statistics reporting (* for $p < 0.05$, ** for $p < 0.01$, *** for $p < 0.001$, **** for $p < 0.0001$). Please indicate which columns are being compared for each statistic.

Response: Thanks for your suggestion. Data for each group (Figures 2-5) were analyzed for statistical significance in antibody abundance at different sites using a

one-way ANOVA (Tukey's multiple comparisons test). Significant differences were determined using the letter marking method. Within each group, different lowercase letters denote statistically significant differences between pairwise samples ($P < 0.05$), while identical lowercase letters indicate no significant difference ($P > 0.05$). After performing pairwise comparisons of antibody abundance at different sites, the differences are indicated by asterisks (see Figure A below), which appears less favorable for observation compared to using letter annotations (see Figure B below). The letter marking method has also been applied in some papers, as seen in Figure C-D below (Zhang L. Nat Commun. 2022; Yang M. Front Nutr. 2022).

A

B

C

D

It would be helpful to include the site specificities beside the representative biotin mAbs in figures 2-4 to ease reader interpretation.

Response: Thanks for your valuable advice. Relevant modification was conducted in figures 2-4.

Please provide a color legend in figure 6.

The data for distinct strains should be separated or labeled in figure 6, as you've just demonstrated that site-specific responses differ across these topotypes. For example, color the O/Mya/98 and O/Tibet/99 data points differently or calculate correlation for each strain.

Response: Thanks for your valuable advice. We have labeled the O/Mya/98 and O/Tibet/99 data points using different colors and calculated the correlation coefficient (r-values) separately. See figure 6.

Reviewer #2 (Comments for the Author):

The author's present work characterizing the neutralizing epitopes from different FMDV O topotypes.

Minor issues

Line 271 Change "None" to "No"

Response: Thanks for your suggestion. We changed it following your advice. See line 272.

Line 303 Pearson

Response: Thanks for your suggestion. We changed it following your advice. See line 305.

Line 332 Change "aroused" to "inferred"

Response: Thanks for your suggestion. We changed it following your advice. See line 335.

Line 336 Change "despite they bind to different antigenic sites" to despite binding to different antigenic sites"

Response: Thanks for your suggestion. We changed it following your advice. See line 337-339.

Line 397-401 These lines do not really add anything in the discussion and could be removed.

Response: Thanks for your suggestion. We have deleted lines 397-401.

Line 395 The differences in NAbs production in cattle among four strains of serotype O viruses were less than that in sheep and pigs indicating the high efficiency of humoral immune responses in cattle.

I do not believe that the efficiency of humoral immune responses in cattle are really different compared to sheep and pigs. For some reason with FMD pigs are more difficult to induce protective immunity compared to cattle. The reason for this is not really known. Discuss this and add a reference.

Response: Thanks for your suggestion. We reopened the discussion as follows: "Studies had shown that although certain high-quality commercial FMD vaccines elicited robust immune protection in cattle, they were not always effective at inducing adequate immune responses in pigs, suggesting significant differences in the immune mechanisms to FMDV between these two species (Stenfeldt C. Front Vet Sci. 2016; Lyons NA. Front Vet Sci. 2016; Martínez JL. Vaccine. 2024). Specifically, cattle typically exhibited a more pronounced immune memory response following FMDV

infection, and their immune systems tended to generate long-lasting neutralizing antibodies, thus providing rapid and sustained protection against reinfection (Juleff N. J Virol. 2009; Barnett PV. Vaccine. 2002). In contrast, pigs exhibited comparatively weaker immune memory responses, with slower kinetics and reduced persistence of neutralizing antibody production (Toka FN. Immunol Lett. 2013).” See line 391-398.

Re: Spectrum03344-24R1 (**Remapping the spatial distribution of neutralizing sites and their immunodominance on the capsid of different topotypes of FMDV serotype O by site-directed competitive ELISA for detection of neutralizing antibodies**)

Dear Dr. Zengjun Lu:

Your manuscript has been accepted, and I am forwarding it to the ASM production staff for publication. Your paper will first be checked to make sure all elements meet the technical requirements. ASM staff will contact you if anything needs to be revised before copyediting and production can begin. Otherwise, you will be notified when your proofs are ready to be viewed.

Sincerely,
Gabriel Parra
Editor
Microbiology Spectrum

Reviewer #1 (Comments for the Author):

My comments have been addressed in the revised manuscript, thank you.

Reviewer #2 (Comments for the Author):

The authors have sufficiently responded to the reviewers comments.